# Organic Amendments and *Trichoderma* Change the Rhizosphere Microbiome and Improve Cucumber Yield and *Fusarium* Suppression

**DOI:** 10.3390/plants14233660

**Published:** 2025-12-01

**Authors:** Yuanming Wang, Xinnan Hang, Cheng Shao, Zhiying Zhang, Sai Guo, Rong Li, Qirong Shen

**Affiliations:** 1Jiangsu Provincial Key Lab of Solid Organic Waste Utilization, Jiangsu Collaborative Innovation Center of Solid Organic Wastes, Educational Ministry Engineering Center of Resource-Saving Fertilizers, Nanjing Agricultural University, Nanjing 210095, China; wym930218@163.com (Y.W.); 2022000110@jou.edu.cn (X.H.); methysc@163.com (C.S.); zhangzhiying_njau@163.com (Z.Z.); lirong@njau.edu.cn (R.L.); qirongshen@njau.edu.cn (Q.S.); 2College of Resources and Environmental Sciences, Nanjing Agricultural University, Nanjing 210095, China; 3The Sanya Institute of Nanjing Agricultural University, Sanya 572025, China

**Keywords:** chitin, straw-waste, *Trichoderma*, cucumber yield, *Fusarium* suppression, rhizosphere metagenomics, soil amendments

## Abstract

Conventional chemical-based control methods for soil-borne diseases often degrade soil quality. The recycling of organic wastes offers a promising solution to simultaneously alleviate environmental pollution and restore soil health. As a beneficial fungus, *Trichoderma* plays a crucial role in enhancing plant performance. However, knowledge of the mechanisms through which organic wastes and *Trichoderma* interact to influence plant performance remains limited. We investigated how the combined application of organic wastes (chitin and straw) and a biocontrol fungus (*Trichoderma*) influenced the rhizosphere microbiome to improve plant performance. Compared with the control, organic waste alone, and *Trichoderma* alone treatments, the combined application of organic wastes and *Trichoderma* significantly (*p* < 0.05) increased cucumber yield and reduced pathogen density. Increased yield and reduced pathogen density were associated with changes in bacterial and fungal communities induced by this combined application treatment. Indeed, this combined application treatment enabled plants to recruit certain potentially beneficial core bacterial (e.g., *Streptomyces* and *Flavisolibacter*) and fungal taxa (e.g., *Trichoderma*), increasing their positive interactions in the rhizosphere. We demonstrate that the combined application of organic wastes and *Trichoderma* can shape distinct rhizosphere bacterial and fungal communities, promoting an increase in beneficial microorganisms and their positive interactions, which contribute to enhanced plant performance.

## 1. Introduction

Cucumber (*Cucumis sativus* L.) is a globally important vegetable crop, and its yield and quality directly influence agricultural productivity and food supply stability [1]. However, Fusarium wilt, caused by *Fusarium oxysporum* f. sp. *cucumerinum*, is among the most destructive soil-borne diseases that threaten cucumber cultivation [2]. Pathogens invade the vascular system, leading to vascular occlusion, wilting, and ultimately plant death, frequently causing yield losses of 30–70% in heavily infested fields and, in severe cases, resulting in complete crop failure [3,4]. Conventional control methods, such as the prolonged use of chemical fungicides and fertilizers, can offer short-term benefits but often lead to soil degradation, reduced microbial diversity, and the development of pathogen resistance [5,6]. Given the growing demand for ecological sustainability and food safety, the development of environmentally friendly and efficient strategies to control Fusarium wilt and simultaneously improve cucumber productivity is urgently needed [7].

Recycling of organic wastes provides a promising approach to simultaneously mitigate environmental pollution and restore soil health [8]. China, as one of the world’s largest agricultural producers, generates approximately 1.04 × 10^9^ tons of crop straw annually [9,10], accounting for a large proportion of global agricultural waste. Traditional disposal methods, such as open burning, not only result in resource waste but also significantly contribute to air pollution [11]. Moreover, the seafood processing industry generates significant amounts of shrimp and crab shell waste, which is rich in chitin—a natural polysaccharide known for its biocompatibility and antimicrobial properties [12]. Owing to its resistance to degradation, the accumulation of chitin can pose significant environmental challenges if it is left untreated [13]. Transforming these wastes into soil amendments can therefore serve two purposes: mitigating environmental burdens and suppressing soil-borne pathogens [14].

*Trichoderma*, a widely recognized biocontrol fungus, is extensively utilized in agriculture because of its diverse roles in disease suppression and plant growth promotion [15,16]. This biocontrol fungus provides superior disease suppression and yield improvement by stimulating beneficial soil microbes and strengthening microbial interactions [17]. *Trichoderma* effectively controls disease and improves plant growth through multiple coordinated mechanisms, such as direct mycoparasitism by recognizing, attaching to, and penetrating pathogenic hyphae to degrade their cellular contents; competition for nutrients, space, and root colonization sites via rapid growth; secretion of hydrolytic enzymes that break down fungal cell walls; and induction of systemic resistance (ISR) in host plants through root colonization, priming the plant’s immune system for enhanced defense and long-term, broad-spectrum protection [17]. Rhizosphere microbial communities play a central role in regulating plant health, and accumulating evidence indicates that the suppression of soil-borne diseases and the promotion of plant growth are strongly associated with shifts in microbial community composition [18,19]. The enrichment of specific rhizosphere bacterial genera and fungal taxa has been linked to a reduced abundance of pathogens and increased crop yields [19]. Chitin acts as a carbon and nitrogen source for *Trichoderma*, which effectively breaks it down through chitinase enzymes to fuel targeted growth [20,21]. Straw provides a sustained supply of organic carbon and serves as a microbial habitat, supporting the rapid proliferation of soil fungal communities [22]. Therefore, chitin and straw may establish a favorable microenvironment, enhancing root colonization and the biocontrol efficacy of the biocontrol fungus *Trichoderma*. However, knowledge of the specific mechanisms through which organic wastes and *Trichoderma* interact to influence plant health remains limited.

To investigate the assembly of rhizosphere microorganisms in plants under the combined application of organic waste and *Trichoderma* and its association with improved plant performance, we conducted a field experiment with the following treatments: control, chitin alone, straw alone, *Trichoderma* alone, and the combined application of chitin, straw, and *Trichoderma*. We examined rhizosphere bacterial and fungal communities across different treatments using Illumina amplicon sequencing. We hypothesized that (1) the increased cucumber yield and reduced pathogen density were associated with changes in bacterial and fungal communities induced by the combined application of chitin, straw, and *Trichoderma* and that (2) core rhizosphere bacteria and fungi recruited by plant roots through the combined application of chitin, straw, and *Trichoderma* would enhance plant performance.

## 2. Results

### 2.1. Sequencing Data

After the removal of ambiguous, short, and low quality reads and singleton OTUs, a total of 5,329,523 bacterial 16S rRNA genes and 5,111,804 ITS high-quality sequences remained for community analyses from rhizosphere soil samples. The number of passing sequences per sample varied from 78,007 to 172,059, with an average of 118,434 for bacteria and from 65,302 to 257,054, and with an average of 150,347 for fungi.

### 2.2. Effects of the Combined Application of Chitin, Straw, and Trichoderma on Crop Yield and Pathogen Density

Before applying different treatments, we quantified the pathogen (*F. oxysporum*) density of the original soils in each treatment, and the results showed that no statistically significant differences in pathogen density were observed among the treatments (Duncan test, *p* > 0.05; Appendix A).

After applying different treatments, the combined application of chitin, straw, and *Trichoderma* (CST) achieved significantly higher cucumber yield and lower pathogen density compared to the control treatment (CK, yield: 16.91%, pathogen density: 11.95%) and the individual applications of straw (S, yield: 12.12%, pathogen density: 11.34%), chitin (C, yield: 15.11%, pathogen density: 9.16%), and *Trichoderma* (T, yield: 15.40%, pathogen density: 8.13%) (Duncan test, *p* < 0.05), while no statistically significant differences in yield or pathogen density were observed among S, C, T, and CK (Duncan test, *p* > 0.05; Figure 1A,B). Moreover, cucumber yield was negatively correlated with *F. oxysporum* density (Spearman correlation: r = −0.72; *p* < 0.05; Figure 1C).

### 2.3. Effects of the Combined Application of Chitin, Straw, and Trichoderma on Rhizosphere Microbial α Diversity and Community Composition

No statistically significant differences in bacterial or fungal diversity (Shannon index) or evenness (Shannoneven index) were observed among the CK, S, C, T and CST treatments (Duncan test, *p* < 0.05; Appendix A). No significant correlations were detected between cucumber yield and rhizosphere microbial diversity (Shannon index) or evenness (Shannoneven index) (Spearman correlation, *p* > 0.05; Appendix A). No significant correlations were detected between pathogen density and rhizosphere microbial diversity (Shannon index) or evenness (Shannoneven index) (Spearman correlation, *p* > 0.05; Appendix A).

Microbial α-diversity (richness/evenness) did not change, whereas β-diversity (community composition) differed significantly, especially for CST. The results showed that the C, S, T, and CST treatments significantly altered the composition of the rhizosphere bacterial and fungal communities compared to the CK treatment (ANOSIM, *p* < 0.05; Figure 2; Appendix A). Moreover, compared with the C, S, and T treatments, CST treatment induced additional modifications to the rhizosphere bacterial and fungal community composition (ANOSIM, *p* < 0.05; Figure 2 and Appendix A). Moreover, the rhizosphere bacterial and fungal community compositions were positively correlated with cucumber yield and negatively correlated with *F. oxysporum* density (Spearman correlation, *p* < 0.05; Appendix A).

### 2.4. Effects of the Combined Application of Chitin, Straw, and Trichoderma on the Rhizosphere Microbial Taxonomic Composition and Its Relationships with Crop Yield and Pathogen Density

Based on the OTU level, the relative abundances of 37 bacterial OTUs, including *Niabella*, *Streptacidiphilus*, *Parasegetibacter*, *Rathayibacter*, *Amnibacterium*, *Tepidibacillus*, *Pedobacter*, *Nakamurella*, *Lysinibacillus*, *Phenylobacterium*, *Sphaerobacter*, *Tumebacillus*, *Streptomyces*, *Propionicicella*, *Nocardia*, *Nakamurella*, *Ornithinibacillus*, *Fulvimonas*, *Pallidibacillus*, *Cohnella*, *Candidimonas*, *Gp3*, *Sphingomonas*, *Bacillus*, *Camellibacillus*, and *Flavisolibacter*, were positively correlated with cucumber yield (Spearman correlation, *p* < 0.05; Appendix A). Based on the corrections, these bacteria were identified as potentially growth-promoting bacteria. The relative abundances of 21 bacterial OTUs, including *Bacillus*, *Altererythrobacter*, *Streptacidiphilus*, *Halobacillus*, *Falsiroseomonas*, *Rhodanobacter*, *Niabella*, *Streptomyces*, *Acidovorax*, *Amnibacterium*, *Cerasibacillus*, *Dyella*, *Chitinophaga*, *Rathayibacter*, *Actinocatenispora*, *Nocardia*, *Pseudalkalibacillus and Symbioplanes*, were negatively correlated with *F. oxysporum* density (Spearman correlation, *p* < 0.05; Appendix A). Based on the corrections, these bacteria were identified as potentially pathogen-suppressing bacteria.

Further Venn diagram analysis was performed on bacterial taxa associated with plant growth promotion and pathogen suppression (Figure 3A). Nine bacterial OTUs were consistently identified as overlapping key taxa that contribute to both potentially growth-promoting and pathogen-suppressive functions in plants, including OTU13549 (*Bacillus*), OTU12 (*Streptacidiphilus*), OTU433 (*Falsiroseomonas*), OTU6610 (*Niabella*), OTU1025 (*Streptomyces*), OTU4454 (*Amnibacterium*), OTU855 (*Flavisolibacter*), OTU2398 (*Rathayibacter*), and OTU312 (*Nocardia*). Among the 9 bacterial OTUs, CST treatment significantly increased the relative abundances of bacterial OTU433 (*Streptomyces*), OTU855 (*Flavisolibacter*), and OTU1025 (*Streptomyces*) compared with those in the CK, C, S, and T treatments (Duncan test, *p* < 0.05; Figure 4A). These bacterial taxa were further designated in this study as core bacteria (which play dual functional roles in potentially promoting plant growth and suppressing pathogens and are induced by the combined application of chitin, straw, and *Trichoderma*).

Based on the OTU level, the relative abundances of 6 fungal OTUs, including *Trichoderma*, *Mortierella*, *Thoreauomyces*, *and Penicillium*, were positively correlated with cucumber yield (Spearman correlation, *p* < 0.05; Appendix A). Based on the corrections, these fungi were identified as potentially growth-promoting fungi. The relative abundances of 8 fungal OTUs, including *Mortierella*, *Trichoderma*, *Corynespora*, *Kochiomyces*, *Thoreauomyces*, *and Rhizophlyctis*, were negatively correlated with *F. oxysporum* density (Spearman correlation, *p* < 0.05; Appendix A). Based on the corrections, these fungi were identified as potentially pathogen-suppressing fungi.

Further Venn diagram analysis was performed on fungal taxa associated with plant growth promotion and pathogen suppression (Figure 3B). Five fungal OTUs were consistently identified as overlapping key taxa that contribute to both potentially growth-promoting and pathogen-suppressive functions in plants: OTU313 (*Trichoderma*), OTU17 (*Thoreauomyces*), OTU46 (*Trichoderma*), OTU1067 (*Mortierella*), and OTU8 (*Mortierella*). Among the 5 fungal OTUs, CST treatment significantly increased the relative abundance of fungal OTU313 (*Trichoderma*) compared with those in the CK, C, S, and T treatments (Duncan test, *p* < 0.05; Figure 4B). This fungal taxon was designated in this study as core fungi (which play dual functional roles in potentially promoting plant growth and suppressing pathogens and are induced by the combined application of chitin, straw, and *Trichoderma*).

Given these results, we used subsequent network analyses to explore associations among the previously defined core taxa.

### 2.5. Relationships Between Core Rhizosphere Bacteria and Fungi with Combined Potentially Growth-Promoting and Pathogen-Suppressive Functions in Plants

Network analyses showed that the relative abundances of the core rhizosphere bacteria OTU433 (*Streptomyces*) (Spearman correlation, r = 0.483, *p* = 0.001), OTU855 (*Flavisolibacter*) (Spearman correlation, r = 0.517, *p* = 0.001), and OTU1025 (*Streptomyces*) (Spearman correlation, r = 0.530, *p* = 0.001) were positively correlated with the relative abundance of the core rhizosphere fungus OTU313 (*Trichoderma*) (Spearman correlation, *p* < 0.05; Figure 5).

## 3. Discussion

We demonstrate here that the combined application of organic wastes and the *Trichoderma* strain (*T. guizhouense* NJAU4742) can shape distinct rhizosphere bacterial and fungal communities, promoting an increase in beneficial microorganisms and their positive interactions, which contribute to enhanced plant growth and pathogen suppression.

Our findings indicate that the combined application of chitin, straw, and the *Trichoderma* strain (*T. guizhouense* NJAU4742) under field conditions effectively promotes plant growth and suppresses pathogens. These results align with previous research, which has shown that the decomposition of organic wastes such as straw and chitin releases nutrients that benefit beneficial soil microorganisms and plant growth [23]. Additionally, previous studies have demonstrated that *Trichoderma* not only promotes growth and controls disease but also strongly degrades organic wastes such as straw by producing various enzymes that facilitate waste decomposition in soil environments [15,24]. Moreover, principal coordinate analysis (PCoA) and ANOSIM revealed that the combined application significantly altered the composition of the bacterial and fungal communities in the rhizosphere soil, driving functional shifts in the rhizosphere microbiome. Furthermore, the modified rhizosphere microbial communities resulting from the combined application were correlated with both the yield of cucumbers and the density of *F. oxysporum*, demonstrating that the combined of the *Trichoderma* strain (*T. guizhouense* NJAU4742) with chitin and straw under field conditions increases plant growth and suppresses soil-borne diseases [25,26]. These findings underscore that the combined application may exert co-occurring growth-promoting and disease-suppressing effects by modulating the rhizosphere microbial community. However, because our experimental design lacked two-factor combination treatments (e.g., C+S, C+T, S+T), it may not be able to distinguish additive from synergistic interactions among treatments.

We further validate the microecological mechanism of co-occurring growth promotion and disease suppression—the specific bacterial and fungal taxa stimulated by the combined application under field conditions, as well as their coupled relationships with cucumber yield and the abundance of *F. oxysporum*. Our results showed that the combined application of chitin, straw, and the *Trichoderma* strain (*T. guizhouense* NJAU4742) effectively increased the abundance of bacterial taxa such as *Flavisolibacter* and *Streptomyces* and the fungal genus *Trichoderma*. These microorganisms exhibited positive correlations with cucumber yield and negative correlations with the abundance of *F. oxysporum*, as well as evidence of positive interactions among these bacterial and fungal taxa [27]. These findings showed that the combination of chitin, straw, and the *Trichoderma* strain (*T. guizhouense* NJAU4742) was associated with higher relative abundance of key microbial taxa—*Flavisolibacter*, *Streptomyces*, and *Trichoderma* previously reported as beneficial and their positive interactions coinciding with improved plant growth and disease suppression [16,28,29]. Previous research indicates that TgSWO from *T. guizhouense* NJAU4742 can expand root cell walls, thereby enhancing root colonization [30]. In addition, *T. guizhouense* NJAU4742 also can stimulate the beneficial indigenous bacteria and fungi, such as *Massilia* and *Aspergillus* [31,32]. Chitin and straw can serve as carbon and nitrogen sources, improving *Trichoderma* populations and selectively enriching beneficial indigenous bacteria and fungi in soils [33,34,35]. In addition, chitin may activate chitinolytic taxa, fostering dynamic interactions with *Trichoderma* through intricate biochemical signaling [36,37,38]. Thus, the combined application of organic wastes and the *Trichoderma* strain (*T. guizhouense* NJAU4742) may further stimulate the population of *Trichoderma* and other beneficial microorganisms that have a positive interaction with it in the soil, increasing crop yields and reducing the number of pathogenic bacteria. Our study confirms the microecological mechanism underlying the co-occurring promotion of plant growth and suppression of diseases by the combined application of chitin, straw, and the *Trichoderma* strain (*T. guizhouense* NJAU4742). This combination (chitin + straw + *T. guizhouense* NJAU4742) can be considered an integrated organic–biocontrol approach with synergistic potential in future agriculture.

Overall, we demonstrate that the combined application of chitin, straw, and the *Trichoderma* strain (*T. guizhouense* NJAU4742) can shape distinct rhizosphere bacterial and fungal communities in plants, promoting an increase in beneficial microorganisms and their positive interactions, which contribute to increased crop growth and reduced pathogen density. Our research hopefully provides valuable insights into the substantial potential of integrating organic waste with biocontrol microbes to improve plant performance in future sustainable agricultural systems. However, our findings were conducted in a greenhouse facility under field conditions for one season and were based on correlative analyses. Subsequently, large-scale field trials can be carried out and experiments over multiple seasons can be conducted to verify the universality of our conclusions. Microorganisms with potential disease-suppressing and growth-promoting functions should also be isolated to verify their disease-suppressing and growth-promoting capabilities.

## 4. Materials and Methods

### 4.1. Site Description and Experimental Design

The field experiment was carried out in August 2020 in a greenhouse located at the Nanjing Institute of Vegetable and Flower Science, Hengxi Town, Jiangning District, Nanjing, Jiangsu Province, China (31°43′ N, 118°46′ E). The study area is characterized by a subtropical monsoon climate, featuring an annual mean temperature of 15.4 °C and an average annual precipitation of 1106 mm [39]. The soil was classified as yellow loam and exhibited the following initial physicochemical properties: a pH of 7.08, an organic matter content of 28.4 g·kg^−1^, a total nitrogen content of 2.04 g·kg^−1^, an ammonium nitrogen concentration of 52.7 mg·kg^−1^, a nitrate nitrogen concentration of 544 mg·kg^−1^, an available phosphorus (as P_2_O_5_) concentration of 242 mg·kg^−1^, and an available potassium (as K_2_O) concentration of 332 mg·kg^−1^ [40].

A randomized complete block design consisting of five treatments and nine replicates per treatment was employed. Each plot measured 8 m^2^ (4 m × 2 m) and was separated by 50 cm-wide cement barriers to prevent cross-contamination. The treatments consisted of (1) control (CK); (2) chitin (C), applied at 2 g·kg^−1^ dry soil; (3) straw (S), applied at 2 g·kg^−1^ dry soil; (4) the *Trichoderma* strain (*T. guizhouense* NJAU4742) (T), inoculated at 1 × 10^3^ spores·g^−1^ dry soil; and (5) combined treatment (CST), incorporating chitin and straw each at 2 g·kg^−1^ dry soil and the *Trichoderma* strain (*T. guizhouense* NJAU4742) at 1 × 10^3^ spores·g^−1^ dry soil. All amendments were uniformly incorporated into the top 20 cm of soil prior to transplanting. Cucumber seedlings are cultivated in seedling trays filled with nutrient-rich seedling substrate at the greenhouses (daytime: 28 °C, night: 25 °C, all-day average humidity of 50%). Cucumber seedlings at the three-leaf, one-heart stage were transplanted at a spacing of 40 cm × 50 cm, with 40 plants per plot. Standard agronomic practices, such as irrigation and weeding, were consistently implemented throughout the growing season, and chemical fungicides were excluded from the management protocol.

The cucumber cultivar Ningfeng Chunqiu, which is widely cultivated in protected cultivation systems and exhibits moderate resistance to Fusarium wilt, was provided by the Nanjing Vegetable and Flower Science Research Institute. Chitin (purity ≥ 95%) was purchased from Shandong Zhongsheng Marine Technology Co., Ltd. (Weihai, Shandong, China), and sieved through a 20-mesh screen prior to application. Rice straw was collected from Lianyungang, Jiangsu Province, air-dried, pulverized, and similarly sieved to a 20-mesh particle size. The biocontrol strain *Trichoderma guizhouense* NJAU4742, known for its dual ability to promote plant growth and suppress disease [31,41,42], was isolated and maintained at the Jiangsu Key Laboratory of Solid Organic Waste Resource Utilization (College of Resources and Environmental Sciences, Nanjing Agricultural University). The strain was cultured in potato dextrose broth (PDB) at 28 °C under shaking conditions (180 rpm) for 72 h, and the concentrations of the spore suspensions were adjusted to 1 × 10^8^ spores·mL^−1^ before application. Basal fertilization was carried out using agricultural-grade urea (N ≥ 46%), superphosphate (P_2_O_5_ ≥ 12%,), and potassium sulfate (K_2_O ≥ 50%). We applied 260 kg of agricultural-grade urea, 1625 kg of superphosphate, and 240 kg of potassium sulfate per hectare as basal fertilizer before transplanting seedlings. The base fertilizers are produced and packaged by Kingenta Chemical Fertilizer Co., Ltd. (Shenzhen, Guangdong, China).

### 4.2. Cucumber Yield Measurement and Rhizosphere Soil Sample Collection

Since the fruiting period (cucumber plants entered the fruiting stage 55 days after transplantation) best represents the final growth outcome of the plants, we recorded cucumber yield and collected rhizosphere soil samples during this period. At this period, we collected all plants (twenty plants) in each plot (a replicate of a treatment), weighed the cucumber fruits from the plants, and recorded the weights as the yield of each plot. And then, the cumulative yield per plot was converted to yield per hectare.

Meanwhile, ten plant samples of each plot were collected using an “S-shaped” sampling method. A S-shaped sampling method systematically collects samples across large areas by following an S-shaped trajectory, ensuring comprehensive spatial coverage. Widely used in agricultural surveys, this approach provides a more representative and efficient alternative to random or other conventional sampling methods. Ten plant roots of each plot were vigorously shaken to remove loosely bound soil, retaining only a thin layer of approximately 1 mm of soil closely associated with the root surface. These roots, along with the adhering soil, were transferred into a sterile flask containing sterile phosphate-buffered saline (PBS). Using sterile forceps, the roots were thoroughly agitated to dislodge all remaining soil particles from the root surfaces. The resulting soil suspension was then centrifuged at 8000 rpm for 20 min, and the resulting pellet was collected as rhizosphere soil of a replicate of a treatment. 45 rhizosphere soil samples (9 replicates × 5 treatments) were stored at −80 °C for DNA extraction, microbial community analysis, and quantification of the pathogen *F. oxysporum*.

### 4.3. Soil Genomic DNA Extraction, Real-Time PCR, Illumina NovaSeq Sequencing and Bioinformatic Analyses

Genomic DNA was isolated from 0.5 g of frozen soil samples using the DNeasy^®^ PowerMax^®^ Soil Kit (Qiagen, Hilden, Germany) following the manufacturer’s protocol. The concentration and purity of the extracted DNA were assessed using a NanoDrop 2000 spectrophotometer (Thermo Fisher Scientific, Waltham, MA, USA), and only those samples with OD_260_/OD_280_ ratios within the range of 1.8–2.0 were selected for further analysis. Quantification of *F. oxysporum* abundance was carried out via real-time quantitative PCR (qPCR) on a Step One Plus Real-Time PCR System (Applied Biosystems, Foster City, CA, USA). The assay employed the specific primers FocF3 (5′-AAACGAGCCCGCTATTTGAG-3′) and FocR7 (5′-TATTTCCTCCACATTGCCATG-3′) [20]. Each 20 μL sample for qPCR analysis consisted of 10 μL of SYBR Premix Ex Taq II (Takara, Kyoto, Japan), 0.4 μL of each primer (10 μmol·L^−1^), 2 μL of DNA template (10 ng·μL^−1^), and 7.2 μL of sterile distilled water. The amplification program included initial denaturation at 95 °C for 30 s, followed by 40 cycles of 95 °C for 5 s and 60 °C for 30 s, with a subsequent melting curve analysis ranging from 60 to 95 °C. Copy numbers of *F. oxysporum* per gram of dry soil were estimated using a standard curve generated from plasmid DNA containing the target gene sequence. All the samples were run in triplicate to ensure reproducibility.

The V4 region of the bacterial 16S rRNA gene was amplified using the primer pair 515F (5′-GTGCCAGCMGCCGCGGTAA-3′) and 806R (5′-GGACTACHVGGGTWTCTAAT-3′) [43], whereas the fungal ITS1 region was amplified with the primers ITS1F (5′-CTTGGTCATTTAGAGGAAGTAA-3′) and ITS2 (5′-GCTGCGTTCTTCATCGATGC-3′) [44]. Each 25 μL PCR mixture sample consisted of 12.5 μL of 2 × Taq Plus Master Mix (Vazyme, Nanjing, China), 1 μL of each forward and reverse primer (10 μmol·L^−1^), 2 μL of DNA template (10 ng·μL^−1^), and 8.5 μL of sterile distilled water. Amplification was carried out under the following thermal cycling conditions: initial denaturation at 95 °C for 5 min, followed by 30 cycles of 95 °C for 30 s, annealing at 55 °C for 30 s, extension at 72 °C for 45 s, and a final extension at 72 °C for 10 min. Amplified products were examined via 1.5% agarose gel electrophoresis, purified using the AxyPrep DNA Gel Extraction Kit (Axygen, San Jose, CA, USA), and quantified with a Qubit 3.0 fluorometer (Thermo Fisher Scientific, Waltham, MA, USA). Purified amplicons from all the samples were normalized to equal concentrations and pooled together to prepare sequencing libraries, which were subsequently sequenced on an Illumina MiSeq platform with paired-end 250 bp reads (PE250) by Guangzhou Magigene Biotechnology Co., Ltd. (Guangzhou, China).

Raw reads were filtered with Trimmomatic to remove low-quality reads (Q value < 20) and adapter sequences. Paired-end reads were merged using FLASH, and operational taxonomic units (OTUs) were clustered at 97% similarity with USEARCH v11.0 after chimeric removal [45]. Taxonomic annotation was performed with the RDP Naive Bayesian Classifier against the RDP 16S rRNA database [46] for bacteria and the UNITE ITS database for fungi [47].

### 4.4. Network Analyses

Network analyses were used to show the relationships between core rhizosphere bacteria and fungi with combined potentially growth-promoting and pathogen-suppressive functions in plants. Network Analyses was calculated by pairwise Spearman correlations. A pairwise Spearman correlation matrix was calculated with the “corr.test” function in the package “psych” [48] in R and *p*-values were adjusted with the false discovery rate method [49].

### 4.5. Statistical Analyses

The data for alpha and beta diversity was rarefied using the “subsample” command in MOTHUR (v 1.48.0. https://mothur.org/). Alpha diversity indices (Shannon and Shannon evenness) were calculated using MOTHUR (v 1.48.0. https://mothur.org/) [50], whereas beta diversity was evaluated through principal coordinate analysis (PCoA) based on Bray–Curtis dissimilarity distances. The significance of differences among treatments was assessed using ANOSIM [50]. One-way analysis of variance (ANOVA) was conducted in SPSS 26.0, and treatment means were compared using Duncan’s multiple range test at *p* < 0.05. The ANOVA data were assessed for normality and homogeneity of variance, and satisfied the assumptions of normal distribution and homoscedasticity. Spearman correlation analysis was performed using R to examine the relationships between cucumber yield, *F. oxysporum* density, and microbial indices (including microbial alpha and beta diversity and the relative abundances of microbial OTUs). Spearman correlation was calculated with the “corr.test” function in the package “psych” [48] in R and *p*-values were adjusted with the false discovery rate method [49].

## Figures and Tables

**Figure 1 plants-14-03660-f001:**
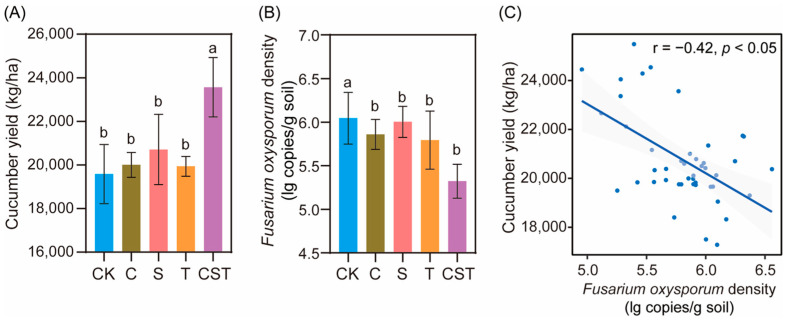
Effects of the addition of different materials on cucumber yield (**A**) and pathogen density (**B**) in the field experiment and the relationship between cucumber yield and pathogen density (**C**). In (**A**,**B**), CK denotes control treatment; C denotes chitin addition treatment; S denotes straw addition treatment; T denotes the *Trichoderma* strain (*T. guizhouense* NJAU4742) addition treatment; CST denotes treatment with a mixture of straw, chitin and the *Trichoderma* strain (*T. guizhouense* NJAU4742). Units for yield are kg/ha; units for *F. oxysporum* density are lg copies/g soil. n = 9 and data are presented as mean ± SD. Statistical significance was calculated by Duncan’s test. Different letters represent a significant difference at *p* < 0.05 according to Duncan’s test. In (**C**), statistical significance was calculated by Spearman correlation.

**Figure 2 plants-14-03660-f002:**
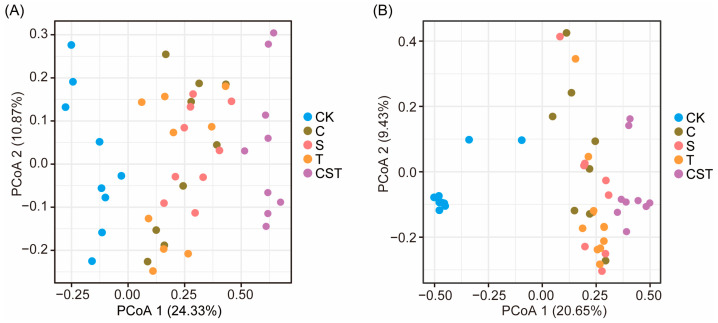
Effects of the addition of different materials on the rhizosphere bacterial (**A**) and fungal (**B**) community composition. CK denotes control treatment; C denotes chitin addition treatment; S denotes straw addition treatment; T denotes the *Trichoderma* strain (*T. guizhouense* NJAU4742) addition treatment; CST denotes treatment with a mixture of straw, chitin and the *Trichoderma* strain (*T. guizhouense* NJAU4742). n = 9 and statistical significance was calculated by ANOSIM.

**Figure 3 plants-14-03660-f003:**
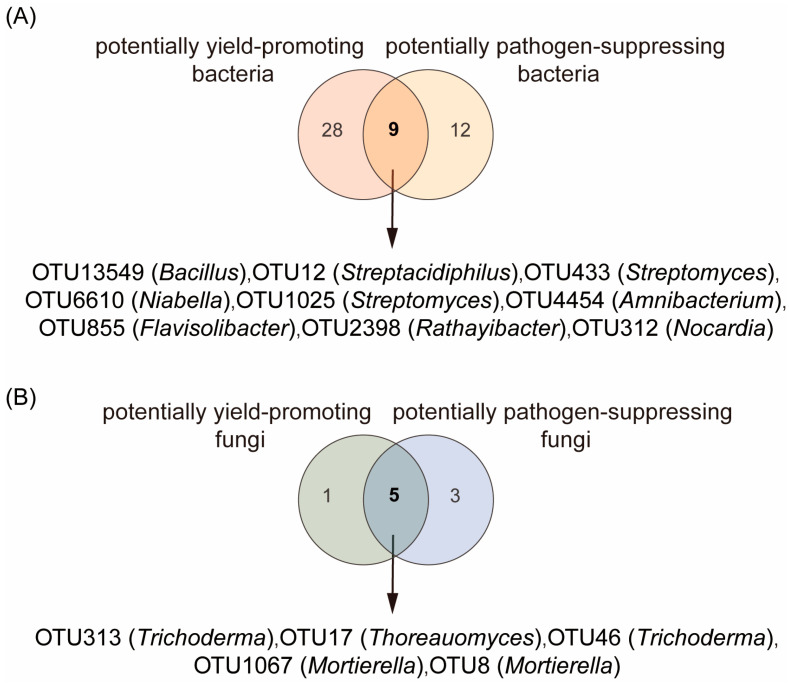
Rhizosphere bacterial (**A**) and fungal (**B**) taxa with both potentially yield-promoting and pathogen-suppressing functions.

**Figure 4 plants-14-03660-f004:**
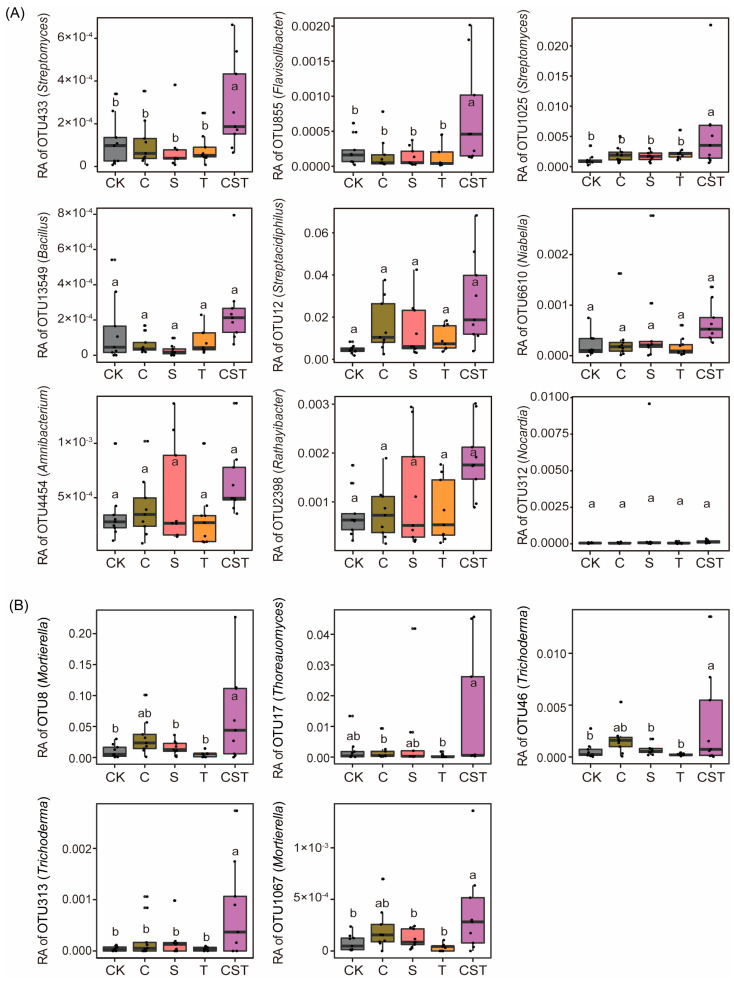
Effects of the addition of different materials on rhizosphere bacterial (**A**) and fungal (**B**) taxa with both potentially growth-promoting and disease-suppressing functions in plants. In (**A**,**B**), CK denotes control treatment; C denotes chitin addition treatment; S denotes straw addition treatment; T denotes *Trichoderma* addition treatment; CST denotes treatment with a mixture of straw, chitin and the *Trichoderma* strain (*T. guizhouense* NJAU4742). n = 9 and data are presented as mean ± SD. Statistical significance was calculated by Duncan’s test. Different letters represent a significant difference at *p* < 0.05 according to Duncan’s test.

**Figure 5 plants-14-03660-f005:**
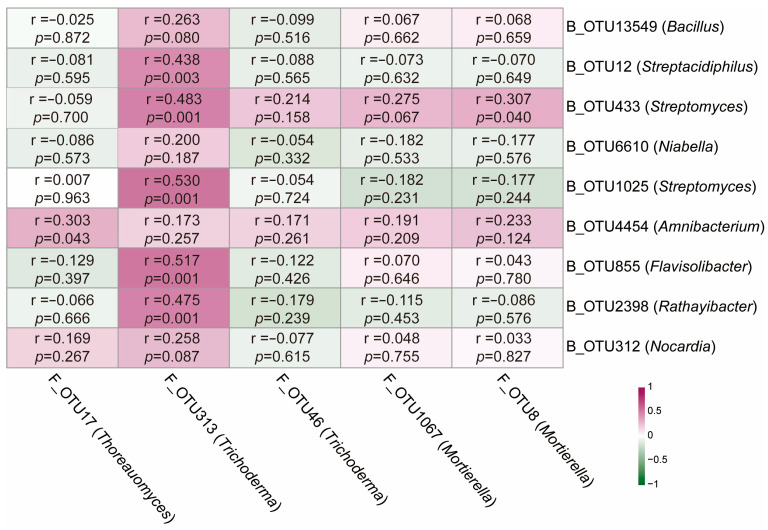
Relationships between core bacterial and fungal groups and both potentially growth-promoting and disease-suppressing functions in plants. B, bacteria; F, fungi. Relationships were calculated by Spearman correlation and *p*-values were adjusted with the false discovery rate method.

## Data Availability

All raw 16S rRNA and ITS gene sequences are available at the CNCB (China National Center of Bioinformation) under the accession number BioProject PRJCA050036.

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
