# Peer review of "Organic Amendments and Trichoderma Change the Rhizosphere Microbiome and Improve Cucumber Yield and Fusarium Suppression"

_plants, 2025, doi:10.3390/plants14233660_

Round 1

Reviewer 1 Report

Comments and Suggestions for Authors

The study by Wang et al. addresses a timely and practical topic in sustainable agriculture, with a clear field experiment design and valuable insights into the interactions between organic waste and biocontrol microbes. I personally found the topic of this work interesting. However, I encourage the authors to make the necessary effort to improve at least along some of the suggested lines below. Below, I outline some minor points that require improvement before publication.

  1. Line 20: The description "Trichoderma plays a crucial role in suppressing plant disease" is overly absolute. Please precise it. I suggest to modify to "Trichoderma is a widely used biocontrol fungus that plays an important role in suppressing plant disease" to align with the diversity of biocontrol microbes in agricultural systems.
  2. Line 36,75: Correct the "im-prove" (hyphen error) to "improve" for consistency in text formatting.
  3. Line 56: When mentioning "China generates approximately 1.04×10⁹ tons of crop straw annually", add a brief note on the data’s core implication (e.g., "accounting for a large proportion of global agricultural waste") to strengthen the rationale for straw recycling.
  4. Line 91: “P”were adjusted, italic? please correct it. Please revise the entire text.
  5. Line105:“pathogen density” which kind of pathogen? please precise it.
  6. Line156:“These bacteria were identified as pathogen-suppressing bacteria” I think this statement is too absolute. You haven't truly verified that the pathogen inhibitory function is merely a correlation. I suggest to modify to " pathogen-suppressing bacteria " to " pathogen-suppressing potential bacteria ".
  7. Line134:“pathogen-suppressing fungi” You haven't truly verified that the pathogen inhibitory function is merely a correlation. I suggest to modify to " pathogen-suppressing fungi " to " potential pathogen-suppressing fungi ".
  8. Line182:please delete the sentence " On the basis of the results of a field experiment” This sentence is redundant.
  9. Lines194,20,209:The genus name of a fungus should be in italics. Please revise the entire text.
  10. Line 187: Please correct it. I suggest to modify to "the microecological mechanism of synergistic growth promotion and disease suppression" to "the microecological mechanism underlying synergistic growth promotion and disease suppression" to conform to academic English expression habits.
  11. Figure legends: Add full names of treatment abbreviations (CK, C, S, T, CST) in the first figure legend (Figure 1) and note "abbreviations apply to all figures" to avoid repeated explanations in subsequent figure legends.
  12. Line 257:"The biocontrol strain Trichoderma guizhouense NJAU4742, known for its dual ability to promote plant growth and suppress disease" Please add appropriate references to enable readers to have a better understanding of this fungi.
  13. Line 327:"…using MOTHUR …" Please add appropriate references.
  14. Line 329:"using ANOSIM" Please add appropriate references.
  15. Line 332:“ oxysporum abundance” Is it abundance or density? Please keep the entire text consistent.

Author Response

Please see the attachment." in the box if you only upload an attachment.

Reviewer 2 Report

Comments and Suggestions for Authors

The manuscript presents an interesting study addressing the combined effects of Trichoderma guizhouense NJAU4742, chitin, and straw on cucumber yield, Fusarium suppression, and soil microbial communities. It’s a relevant topic for sustainable agriculture. Overall, the paper is well organized, and the experiments appear carefully executed. However, some clarification and adjustments are necessary to improve scientific accuracy, data interpretation, and reporting consistency.

The comments below are intended to strengthen the manuscript and help the authors present their results more clearly and rigorously.

Abstract

Lines 22-23 - provide values for better understanding the meaning of significant differences;

Line 23 – replace “influenced” with “associated”  – to avoid stating causality

Introduction

Line 60: Please verify reference [16]. The cited article does not provide evidence that Trichoderma stimulates beneficial organisms, strengthens microbial interactions, or directly suppresses disease. Replace it with a more appropriate reference.

Consider including a brief description of known Trichoderma biocontrol mechanisms (mycoparasitism, competition, extracellular enzyme secretion, induced systemic resistance, ...).

Briefly mention the individual benefits of chitin (substrate for Trichoderma colonization) and straw (organic carbon source, microbial niche…).

Please explain in the introduction why the combination of chitin + straw + Trichoderma is expected to have synergistic effects.

Results:

section 2.1.

Lines 82–89, two concerns:

-The same idea is repeated three times. Please rewrite and standardize notation (use “Figure 1A,B” consistently). One concise sentence summarizing the comparisons (CST vs. CK; single treatments vs. CK; CST vs. singles) is enough.

-The text describes statistical significance but not the magnitude of the effect. Please include mean ± SE (or SD) values or percentage changes for yield and Fusarium density in either the text or figure to show effect size, not only significance.

Line 91: R² cannot be negative; likely the correct value is r = –0.72 (Spearman correlation). Please confirm.

Figure 1 legend: Add n = 9 (if correct), describe error bars, units for yield, and statistical tests per panel (e.g., Duncan’s test for Figures 1 and 4; Spearman correlation for Figures 1C + 5). Indicate whether data are presented as mean ± SD or mean ± SE and how significant differences are shown (letters, symbols, etc.).

Replace “influenced by” with “associated with” when linking community composition to yield or Fusarium suppression. Keep causal interpretation for the Discussion.

Confirm whether the field soil was screened for Fusarium before the experiment. If pre-treatment data exist, please include them; if not, acknowledge this as a limitation and describe how spatial heterogeneity was minimized.

Justify the use of PCoA1-only correlations or expand analysis (multivariate fit using multiple axes or other).

Apply multiple testing correction (FDR) to OTU correlation tables (S7–S10) and indicate which taxa remain significant; confirm that the “core” taxa remain robust after correction.

Section 2.2

The text appears contradictory to me because α-diversity results (not significant) are presented alongside significant β-diversity differences without explanation. Please clarify that α-diversity (richness/evenness) did not change, whereas β-diversity (community composition) differed significantly, especially for CST.

Figure 2 legend: Add statistical information (ANOSIM values, P-values, n of replicates).

Section 2.3.

Some text, such as “These bacteria were identified as growth-promoting” and “These fungi were identified as pathogen-suppressing” seem to overstate the evidence. The designations appear to be based on correlations rather than functional validation. Please revise to indicate that these taxa were associated with higher yield or lower pathogen abundance and are potentially growth-promoting or pathogen-suppressing.

please clarify that “core bacteria/fungi” refers to taxa showing both associations, not to the conventional definition of core microbiota.

Figure 4 legend: Missing key information—add n, error description, and significance test…

Section 2.3.

The term “synergistic effects” should be replaced with “co-occurrence” or “positive association” to avoid over-interpretayion of positive correlations.

Figure 5 and Section 2.4 should report correlation coefficients, P-values, and specify whether multiple-test correction was applied.

The text implies microbial “network” analysis, but no such method is described in Methods. If network inference was intended, please describe it; otherwise, refer to these as pairwise correlations.

Clarify the novelty of Section 2.4 compared to 2.3 (it could be “To explore associations among the previously defined core taxa…”). I think it would improve logical flow.

Discussion:

Please ensure that all microbial genus and species names, like TrichodermaMortierellaStreptomycesBacillus are italicized throughout the manuscript.

Additionally, specify the Trichoderma strain (T. guizhouense NJAU4742) consistently.

The first paragraph is repetitive of Section 2.1; please reduce to one concise summary sentence.

The discussion section should acknowledge that associations between microbial taxa and plant performance are correlative, not causal. Please rewrite  “CST increased the abundance of growth-promoting microbes, leading to yield improvement”  to something like:  “CST was associated with higher relative abundance of taxa previously reported as beneficial, coinciding with improved yield.”

Please avoid referring to “synergistic effects” between bacteria and fungi unless experimentally verified. Correlation does not imply interaction. “Co-occurrence” or “positive association” would be more accurate.

The discussion could also better connect the present results with the specific mechanisms known for T. guizhouenseNJAU4742 (e.g., TgSWO-mediated root colonization reported by Meng et al., 2019). This would strengthen the interpretation

The section on practical implications could benefit from quantifying yield improvement and emphasizing that the study was conducted over one season in a greenhouse facility in field conditions (maybe a limitation here)

In this section, the text does not clearly emphasize why this combination (chitin + straw + Trichoderma) is novel. Add a sentence framing it as an integrated organic–biocontrol approach with synergistic potential.

And the manuscript never acknowledges missing baseline Fusarium data, short duration, or correlative nature of findings. A short limitations paragraph would make the paper stronger.

Could explicitly state that organic amendments provide carbon and nitrogen substrates that selectively enrich beneficial taxa.

Avoid using “identified as” or “synergistic” unless mechanistically proven. Replace with “potential,” “associated,” or “co-occurring.”

Methods Section

- Lines 230–237: The manuscript reports temperature, rainfall, and soil composition but does not indicate the source of these data. Were these measured by the authors? If so, add methods; if not, please cite the source.

-consider including germination and seedling conditions (substrate, temperature, fertilization).

-Lines 263–264: Indicate the fertilizer supplier and application rate.

-Line 257: Add a citation when introducing the biocontrol strain T. guizhouense NJAU4742 (“known for its dual ability to promote plant growth and suppress disease”).

-Clarify whether soil was screened and homogenized for Fusarium before treatments to ensure equal initial infestation levels. Without this, results may be influenced by an uneven pathogen distribution, which would be concerning.

-Statistical analysis: Please specify tests for normality and homogeneity of variance before ANOVA, the version of MOTHUR used, and whether data were rarefied for diversity analyses.

Reviewer 3 Report

Comments and Suggestions for Authors

The manuscript addresses a timely and interesting topic: the use of organic soil amendments in cucumber production and their impact on crop performance and soil-borne pathogens (particularly Fusarium). Using metagenomic analysis of rhizosphere communities, the authors evaluate four soil treatments: i) chitin (C); ii) straw (S); iii) Trichoderma (T); and iv) the combination of all three (CST), compared to an untreated control (CK). The manuscript is generally well organized, and the results are clearly presented. However, several critical issues must be addressed because they affect interpretation of the data. Detailed points are the following:

  1. Keywords. Keywords should be terms not already present in the title. I recommend removing words that duplicate the title and replacing them with concise, searchable terms. Suggested keywords:
    chitin, straw-waste, Trichoderma, cucumber yield, Fusarium suppression, rhizosphere metagenomics, soil amendments.
  2. Experimental design, replication and sampling. The manuscript does not clearly state how many plants were harvested per treatment and per replicate for yield assessment (the “S-shaped” sampling is mentioned but not quantified). Please state explicitly, number of plants harvested per plot/replicate, and how plot-level yield was calculated (e.g., mean weight per plant, total plot yield).
    Similarly, for the rhizosphere metagenome work, state how many plants were sampled per treatment and replicate, and whether samples were processed individually or pooled (and if pooled, how many individuals per pool). Clarify the number of biological replicates used in sequencing and the downstream statistics based on those replicates.
  3. Title and scope (“plant performance”). The term “plant performance” in the current title is too general because the manuscript reports only yield (harvested cucumber weight) and pathogen density (Fusarium oxysporum f. sp. cucumerinum) in the rhizosphere. I suggest rephrasing the title to reflect the measured outcomes more precisely. Possible titles:
  1. Organic Amendments and Trichoderma Change the Rhizosphere Microbiome and Improve Cucumber Yield and Fusarium Suppression
  2. Combined Application of Chitin, Straw and Trichoderma Alters the Rhizosphere Microbiome, Increases Cucumber Yield, and Suppresses Fusarium

Choose one that best matches the manuscript emphasis.

  1. Interpretation of combined vs single treatments (additivity vs synergy). Figure 3 and related results show that single-material treatments (C, S, T) have differential, partially overlapping effects on community structure, while the combined CST treatment produces a distinct compositional shift. This could reflect additive effects, true synergy among treatments, or nonlinear interactions. Because the experimental design lacks two-factor combination treatments (e.g., C+S, C+T, S+T), it is impossible to separate additive from synergistic interactions. The authors should:
  1. Acknowledge this limitation explicitly in the Discussion.
  2. Avoid claiming synergy unless supported by factorial data.
  3. Discuss plausible mechanisms by which the combination could produce non-additive effects (e.g., chitin stimulating chitinolytic taxa that interact with Trichoderma or straw-derived carbon), with relevant citations.
  4. State the number of days after transplant when rhizosphere samples were collected and justify why that time point is appropriate (or discuss how temporal dynamics might affect results). Provide citations on temporal dynamics of rhizosphere responses to amendments.
  1. Data presentation and visual clarity.
  1. Figures 1, 2, and 4 show large differences between single-material treatments and CST. Given the missing two-factor combinations, the authors must elaborate and justify possible reasons for the magnitude of the CST effect (biological rationale, literature support).
  2. In Figure 2 the colors for CK and C are very similar and difficult to distinguish, especially given the small point size. Please choose more distinct colors and increase symbol size or use different shapes to improve readability.
  3. In Figure 3, panel labels and legends should be clearer (see next point).
  1. Figure 3 labelling and OTU identification.
  1. Rephrase labels in Figure 3: use “yield-promoting” instead of “positive with yield” and “pathogen suppression” instead of “negative with pathogen.”
  2. If feasible, annotate the most important OTUs with species (or the closest taxonomic assignment) either on the figure or in the legend. At minimum provide the species-level identity for OTUs discussed in the text (or include a supplementary table linking OTU IDs to taxonomy).

In conclusion, the study has potential and presents interesting data, but the manuscript requires major revision to (1) clarify experimental replication and sampling, (2) adjust interpretations about combined-treatment effects to acknowledge the lack of factorial combinations and avoid unsubstantiated claims of synergy, (3) improve figure clarity and labels, and (4) refine title/keywords to match the measured outcomes. After these issues are addressed and the Discussion revised to better contextualize the CST effect, the manuscript will be suitable for further consideration.

Reviewer 4 Report

Comments and Suggestions for Authors

This study investigates the synergistic effects of the combined application of organic wastes (chitin and straw) and a biocontrol fungus (Trichoderma) on cucumber yield, pathogen suppression (Fusarium oxysporum), and rhizosphere microbiome composition. However, the manuscript requires substantial improvement. My major concerns are below

Comments

Methods

It is not clear whether the chitin and paddy straw used in the experiment were pretreated or applied in their raw form. Since raw chitin and raw straw are poorly degradable, their slow decomposition could have limited nutrient release and influenced microbial succession. Please specify the pretreatment status of these materials and discuss how using raw substrates may have affected the results.

A reference is needed for the strain Trichoderma guizhouense NJAU4742 to support its characteristics and previous usage.

Please clarify whether the authors aimed to collect only rhizosphere soil, or whether rhizoplane-associated microbes were also included. If only rhizosphere soil was used for sequencing, then the purpose of the additional vortexing and sonication steps should be explained.

The “S-shaped” sampling method may not be familiar to all readers. Please provide a reference or short explanation.

After the first occurrence, Fusarium oxysporum should be abbreviated as F. oxysporum throughout the manuscript.

The statistical methods are incomplete and need further details, including how assumptions were tested, how multiple comparisons were handled, and whether PERMANOVA or other multivariate tests were considered.

Results

The results section needs substantial improvement for clarity and completeness.

The authors state that paddy straw application increased yield. However, since unprocessed paddy straw is slow to decompose, it is unclear how yield improvement occurred. Additionally, Trichoderma is recognized as a plant growth-promoting (PGP) fungus, yet its individual application did not significantly affect yield. The authors should address this inconsistency and explain why the combined treatment produced a different outcome.

There is no information provided about sequencing read counts, data filtering, or overall richness indices for bacteria and fungi. Basic sequencing statistics must be included. Also Fusarium qPCR results imfromation only compared with yield parameter. Other information were missing.

Sec 2.3 the authors need to clarify how the “functional” roles of the bacterial and fungal taxa were inferred. It is also necessary to specify which taxonomic level (e.g., genus or OTU) was used for this classification and on what basis these functions were assigned

Round 2

Reviewer 3 Report

Comments and Suggestions for Authors

The authors have thoroughly elaborated and adequately addressed all reviewers’ suggestions improving the manuscript for publication.

Reviewer 4 Report

Comments and Suggestions for Authors

Dear Authors,
After reviewing the previous comments, I believe the manuscript has improved, and from my side I have no further remarks to add.